# Estimating the Semantic Density of Visual Media

## ABSTRACT

Image descriptions provide precious information for a myriad of visual media management tasks ranging from image classification to image search. The value of such curated collections comes from their diverse content and their accompanying extensive annotations. Such annotations are typically supplied by communities, where users (often volunteers) curate labels and/or descriptions of images. Supporting users in their quest to increase (overall) description completeness where possible is, therefore, of utmost importance.

In this paper, we introduce the notion of *visual semantic density*, which we define as the amount of information necessary to describe an image comprehensively such that the image content can be accurately inferred from the description. Together with the already existing annotations, this measure can estimate the annotation completeness, helping to identify collection content with missing annotations.

We conduct user experiments to understand how humans perceive visual semantic density in different image collections to identify suitable proxy measures for our notion of visual semantic density. We find that extensive image captions can serve as a proxy to calculate an image's semantic density. Furthermore, we implement a visual semantic density estimator capable of approximating the human perception of the measure. We evaluate the performance of this estimator on several image datasets, concluding that *it is feasible to sort images automatically by their visual semantic density, thereby allowing for the efficient scheduling of annotation tasks*. Consequently, we believe that the visual semantic density estimation process can be used as a completeness measure to give feedback to annotating users in diverse visual content ecosystems, such as Wikimedia Commons.

## CCS CONCEPTS

• **Computing methodologies** → Perception; *Visual content-based indexing and retrieval*; • **Applied computing** → *Annotation*; Image composition; • **Information systems** → *Crowdsourcing*; *Digital libraries and archives*; Web searching and information discovery.

## KEYWORDS

Visual Semantic Density, Visual Perception, Multimodal Document Annotation, Annotation Completeness

**ACM Reference Format:**
Anonymous Author(s). 2024. Estimating the Semantic Density of Visual Media. In *Proceedings of ACM Multimedia Conference 2024 (ACMMM'24)*. ACM, New York, NY, USA, 9 pages. https://doi.org/XXXXXXX.XXXXXXX

## 1 INTRODUCTION

The old adage *"a picture is worth a thousand words"* is often quoted in the context of image classification, labeling, or captioning. While this was clearly meant metaphorically, it stands to reason that images can visually express many and complex concepts. The number of concepts expressed this way is, however, dependent on the image itself.

Image descriptions are used to train and evaluate classification algorithms, augment image search systems, and organize the collections. Therefore, having comprehensive image descriptions is critical for the performance of the systems that use them.

Large-scale image collections, such as what can be found on Flickr,[1] usually describe images by tagging the most visibly prominent entities in the picture. Consequently, as we show in this paper, most of the images in these collections are annotated with only a few labels per image, even when images contain high visual complexity that deserves a larger number of words. There is, thus, a need to devise a measure for image description completeness. *But when are image descriptions actually "sufficiently" comprehensive?*

In this paper, we define the notion of visual semantic density (VSD) as the amount of information necessary to describe an image comprehensively so that the image can be accurately inferred from the description. We aim to measure VSD as an estimate for image annotation completeness.

Such an estimate is useful for many applications. One of the most prominent applications is the case of peer-production image annotations. When these image collections and their corresponding descriptions are created as a result of a peer-production process, such as in Wikimedia Commons, users can extend the annotations curated by other users. This way, the quality of image descriptions can improve by means of collaboration. Given the scale of these image collections, it is crucial to help users identify the images whose description completeness can be improved.

Our end goal is to show this estimate to users to encourage them to improve existing annotations, similarly to the way ReCoin displays the relative completeness of individual Wikidata items, as compared to other entities of the same type [1]. Instead of estimating an image's VSD as an absolute number—which is a highly complex task—we estimate a ranking of images based on their VSD.

Recent AI developments facilitated the implementation of methods that, for example, learn to classify images from text [21], automatically describe images [16], and link images to Web entities [2]. However, to the best of our knowledge, no published work has estimated our notion of VSD.

To this end, we first conduct user experiments using crowdsourcing to identify how humans perceive the notion of VSD. More

---

[1]https://flickr.com/

specifically, we run these experiments using data from widely used large-scale image collections: Wikimedia Commons, YFCC100M, LVIS, Visual Genome, and a combination of Stanford Paragraphs and Localized Narratives. The results of these experiments allow us to determine suitable proxy measures for VSD. From all the compared options, extensive image captions show the highest Spearman's rank correlation ($\rho_s$) with our notion of VSD. Furthermore, we implement an end-to-end VSD estimator that resembles the human perception of the VSD measure. From the estimator evaluation, we conclude that the model is capable of sorting the images based on VSD, as the correlation between the model's output and any of the manual orders that we compute (including a crowd-based one) is similar to the correlation between any of the manual orders.

In summary, the main contributions of this paper are:

(1) We present the notion of visual semantic density, which helps assess the completeness of image descriptions.
(2) We conduct user experiments to capture the human perception of visual semantic density and show that the extent of image captions correlates with visual semantic density.
(3) We present a mechanism for sorting pairs of images based on crowdsourced VSD human assessments, extending a state-of-the-art sorting algorithm.
(4) We provide an end-to-end visual semantic density estimation based on a neural network implementation.

In the following, we provide a brief overview of related work in Section 2 before offering a definition of our notion of visual semantic density in Section 3. Section 4 then investigates which existing annotated image datasets could be repurposed as possible sources of ground truth for the estimation of VSD, after which Section 5 studies how it is perceived by humans. Sections 6 and 7 present mechanisms to estimate the VSD of an image using synthetic image captions or and end-to-end approach, respectively. In Section 8, we discuss our insights before offering some outlook in Section 9.

## 2 RELATED WORK

There is a long tradition in image annotation research in computer vision [23]. One of the tasks in this field consists of annotating a bounding box in an image—the rectangles surrounding an object in an image. Snapper [26] is a tool that assists users in the task of annotating the bounding box by allowing them to snap the area and then automatically modifying the bounding box to fit the object optimally, leading to a reduction of the time that the user needs to spend in the annotation as compared to other approaches for the same task. More recently, fully automated segmentation mechanisms such as Segment Anything [10] have emerged, which require little to no manual input for object segmentation. While having a count of objects is a reasonable basis for measuring visual semantic density, this number alone does not fully capture our notion of visual semantic density.

[2, 8] link images to entities in Wikipedia. Given that Wikipedia is a general knowledge data source with natural language text describing the entities, these links are highly valuable for augmenting visual media browsing and searching over large-scale image collections. However, the text is about the entities instead of the images themselves. Therefore, these kinds of approaches do not help solve our task.

Language models have also been used in the field of visual media management. CLIP [21] uses multimodal embeddings to perform image classification from textual descriptions of images. This method generates new information about the image (i.e., as the authors show in their example, the method can learn that the picture is about a dog when the textual description mentions "Pepper the aussie pop"). While this classification contributes to the notion of visual semantic density, the approach still depends on the input text. Along the same lines, image segmentation algorithms match a text description to an area of an image [14]. [4] allows users to have multimodal prompts, including text and images, with the purpose of, for example, asking the model to describe the image given in the input. This is highly relevant to our work, as it generates an image description. We apply this method in our work, as discussed in Section 6.

## 3 THE NOTION OF VISUAL SEMANTIC DENSITY

We define *visual semantic density* (VSD) as the minimal amount of information necessary to describe an image entirely and uniquely, such that the image content can be inferred from the description. This information, for example, in the form of a natural language image description, should provide sufficient detail to derive the image from it. Titles, alternative texts, and captions provide representative descriptions of images. However, they are commonly short, concise, and lack certain specific details. The words defining the VSD of an image should, e.g., allow an artist to paint the image accurately by reading the words. Hence, most real-world annotations are not at the full VSD, as there is likely to remain some ambiguity.

Figure 1 shows an example image. A possible title for this image would be *"Golden retriever in thoughts"*. In a tagging system, the image could be labeled with 'dog' and 'golden retriever.' An alternative text could be *"A portrait of a golden retriever."*, and a possible caption could be *"A sideways portrait of a golden retriever towards the right."*. However, based on our definition of VSD, we would expect the comprehensive description to be, for example: *"a sideways portrait of a golden retriever, including the head and the bust; dog looking straight but picture taken from the right side of the dog: light coming from below; right ear slightly up and slightly blurred as if the picture was taken when someone was holding the ear up; bright brown eyes having some light reflection; black nose; four long blonde moustache hairs; with a person, possibly a woman, in the background in the blurred or bokeh part of the picture with the eyebrows, eyes, and nose visible; picture taking with wide or medium aperture; a wall in the blurred part of the picture, from which two thirds is marble-like (mostly white and with some spots), and the rest is painted in mustard yellow except a vertical rectangle rotated around 45 degrees from which we see four fifths; in the yellow part there is the bottom of a frame with a picture, in the marble part there is a circle, and a small black rectangle with a white mark in front that looks like toilet paper."* With this level of detail, it is more likely to derive the image from words.

In psychology *visual density* is defined as the "number of visual elements in the unit area of visual design" [5, 22, 27]. This definition, however, is insufficient in our scenario, as one could say that the

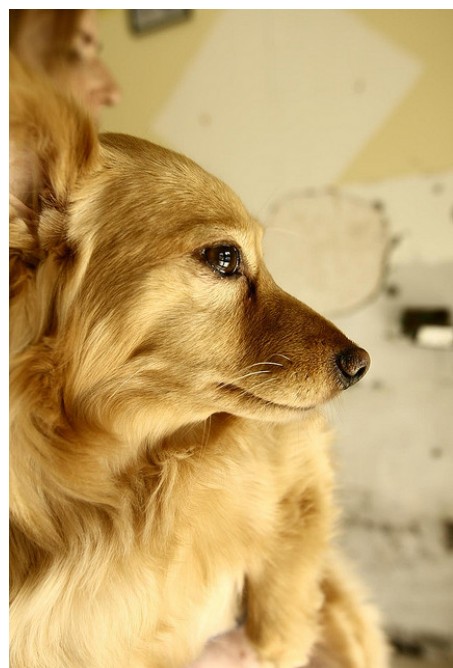

**Figure 1: Example image showing a golden retriever. We use this image as an example to illustrate our notion of visual semantic density. The image is sourced from the LVIS dataset.**

elements are 'dog,' 'woman,' 'wall', and 'toilet paper.' This set of elements lacks descriptive power.

Based on feedback received in our preliminary experiments (Section 5.1) and our observations when conducting exploratory tasks, we identify the following properties of our VSD concept:

(1) *VSD depends on the elements in the image, their diversity, and the characteristics of the image.* Colourfulness, sharpness, light, the presence of expressions, action, and details all influence VSD.
(2) *The salience of objects is relative to the other objects in the image.* Certain elements may be considered when eliciting the description of some images and ignored in others.
(3) *Context influences VSD.* When the VSD assessment is done in pairs of images, the *perception* of the visual semantic density of an image may change depending on the image it is compared to.
(4) *Background knowledge may influence the assessment of the VSD of an image.* Familiarity with the image content may lead to shorter descriptions, due to the use of a denser vocabulary, or longer ones, due to the recognition of additional relevant elements.
(5) *The VSD of a set of images is, at most, the sum of the individual VSDs.* If we measure the VSD of $n$ images in a collage, the total visual semantic density will be at most of the sum of the individual measurements.

Any measure for VSD should account for these properties.

# 4  POSSIBLE SOURCES OF GROUND TRUTH

Since no ground-truth dataset annotating the semantic density of visual media exists, we investigate the applicability of other existing datasets to our task. To this end, we consider three different types of annotations: user-generated tags, object-detection masks, and detailed image captions.

## 4.1  Community-generated Tags

The Wikimedia Commons dataset[2] consists of 104,418,660 media files as of the 29th of March 2024, including images, audio files, and videos, all licensed under free licenses.[3] In the context of this paper, we focus exclusively on the images available. These images represent "photographs, diagrams, drawings, paintings, animations, maps, and symbols." [4]

Wikimedia Commons is an evolving dataset curated and maintained by volunteers who upload and describe the media objects via the project wiki. The description of the file's content can be edited by the file's author and complemented by anyone registered in Commons, promoting collaboration among community members.

While tools exist that help editors add labels (e.g., to add geolocation information or classify images), the labeling effort is primarily human-based. Analogously to the process in other Wikimedia projects, such as Wikipedia and Wikidata, community editors monitor changes implemented in the dataset, which can be reverted if they result from vandalism.

As a result of the integration between Commons and Wikidata [25]—the multilingual free knowledge base containing structured data about people, places, events, and other types of items—the labels in Commons represent Wikidata items. For instance, a picture showing the opera building in Melbourne can have the label Q3141, which is the item in Wikidata for the city of Melbourne, described with geographical information, population, inception information, government information, and links to further databases. The Wikimedia Commons SPARQL endpoint[5] exposes the structured data image labels, enabling federated queries with Wikidata.

The Yahoo Flickr Creative Commons 100 million (YFCC100M) collection [24] consists of 100 million images and videos collected from Flickr. In addition to the actual images and videos, the dataset contains all the metadata for each element of the dataset. This metadata also includes user-generated tags associated with each image. In contrast to Wikimedia Commons, these tags are generated exclusively by the person who posted the image to the platform. There is, therefore, a different relation between images and tagging users, which is why we would also expect a different distribution of tags compared to Wikimedia Commons.

For both collections, we use the raw tag count to estimate an image's semantic density.

## 4.2  Object detection annotations

Another possible source for estimating semantic density comes from datasets that explicitly and densely annotate objects in images. For

---

[2]Wikimedia Commons https://commons.wikimedia.org/wiki/Commons:Database_download
[3]Free Licenses https://freedomdefined.org/Licenses
[4]https://commons.wikimedia.org/wiki/Main_Page
[5]Wikimedia Commons SPARQL Endpoint https://commons.wikimedia.org/wiki/Commons:SPARQL_query_service

our experiment, we consider two such datasets, Visual Genome [12] and LVIS [6].

The Visual Genome dataset consists of images, each with a series of bounding-box annotations for visible objects that were drawn by humans. The object classes are mapped to Wordnet. The annotations not only include the localized objects inside the bounding boxes, but also describe their semantic relations in a graph structure, capturing the semantics of the image in a scene graph.

LVIS is an object recognition dataset with fine-grained object polygon annotations. It is based on MS COCO [15], extending the object annotations by an additional roughly 1200 different object classes.

For both of these datasets, we use the number of object annotations per image as a proxy for its semantic density.

### 4.3 Detailed image captions

The third possible source for estimating VSD is obtained by looking at the length and diversity of the text needed to describe an image. For this, we combine two datasets with detailed image captions: Stanford Paragraphs [11] and Localized Narratives [20]. While Stanford Paragraphs uses images from Visual Genome, Localized Narratives uses Open Images [13], MS COCO [15], Flickr30k [28], and ADE20k [29] as image sources. Both datasets then augment these images with detailed textual descriptions of their content annotated by humans.

In order to obtain an estimated semantic density of the images in these datasets, we process their captions using Core NLP [19]. We extract adjectives, adverbs, nouns, and verbs for each caption and use the size of the set of their lemmatized versions as a proxy value.

## 5 HUMAN PERCEPTION OF VSD

Having defined our notion of VSD and identified some possible proxy measures, we next investigate how humans perceive this notion. In a series of experiments with human participants, we collect information about their perception of the semantic density of selected images.

### 5.1 Preliminary experiments

We defined a VSD assessment task as a survey in Qualtrics,[6] where, given a collection of pairs of images, users are asked to select the image with the highest VSD. We chose a design with pairs of images because there is evidence suggesting that pairwise comparisons are faster and more effective than ranking or individual ratings [3]. For each pair, users could choose among four options: i) the first image has the highest VSD, ii) the second image has the highest VSD, iii) both images have an equal VSD, iv) the user cannot really decide.

The goal of these preliminary experiments was to measure the extent to which different people agree on the assessment of VSD. To test task clarity and estimate completion time, we first ran a pilot experiment with six scientific researchers in our research lab to accomplish the task. As a dataset, we used 30 pairs of images from Stanford Paragraphs. We selected these pairs of images randomly following a bucketing system based on the number of labels per image available in the original dataset, which represents images and

labels collected from Flickr. Based on the distribution of the number of labels per image, we split the data into three groups: low (images with less than 20 labels), medium (images with between 20 and 60 labels), and high (images with more than 60 labels), assuming that the more labels, the higher the VSD. An illustration of the label distribution for all the datasets is shown in the supplementary materials. We randomly sampled ten pairs of images from each group and built pairs that contained intra-class images (i.e., low-low, medium-medium, high-high) and pairs that contained inter-class images. Moreover, we replicated every pair, swapping the order of the images to be able to evaluate the extent to which users are self-consistent.

Following the same procedure, we ran further pilot experiments in Prolific.[7] We published one task with 60 pairs of images from the LVIS dataset and one task with 60 pairs of images from the Visual Genome dataset. We collected responses from ten different Prolific workers and ensured that there was an empty intersection between the sets of workers of the two tasks. We set screeners for workers whose primary language is English, with an approval rate of a minimum of 90%, and at least 50 previous submissions. We extended the task by adding two pairs of images as attention checks. We configured the Prolific task workflow such that participants who did not answer the attention checks correctly could not proceed with the task. At the end of the task, we asked participants to describe how they interpreted the concept of VSD throughout the task, and we also provided an open-ended text field to collect their feedback on their task.

These preliminary experiments helped us iterate over the design of our human rating task. The initial task version, tested by our research lab colleagues, framed the instructions referring to VSD as the number of elements in the image. Participants indicated that other factors influenced their assessment. Hence, we refined the instructions in subsequent versions of the task, mentioning that this notion is defined by the number of words needed to describe the image comprehensively.

Furthermore, we observed low inter-rater agreement as measured by Fleiss' Kappa, both in intra- and inter-class tasks for LVIS and Visual Genome ranging from $-0.031$ to $0.448$. Moreover, we identified that self-consistency was far from perfect in the Prolific experiments. Participants indicated contradictory answers in inverse pairs (i.e., they did not choose the same image when selecting the image that had the highest VSD). When we count the number cases in which users selected opposite answers for inverse pairs (i.e., they selected different images as the ones with highest VSD, or in one of the cases they indicated that the images had equal VSD), in the LVIS Prolific experiment, we obtained $\mu = 4.00$ and $\sigma = 3.09$. In the case of the Visual Genome dataset, we obtained $\mu = 3.40$ and $\sigma = 3.06$. The maximum number of possible inconsistencies here is 30, as there were 60 pairs. This shows that the notion of VSD is complex, and people are sometimes conflicted about it.

### 5.2 Human-in-the-Loop Sort

To get an estimate on how well our different possible ground truth sources introduced in Section 4 align with our notion of VSD, we need a way of sorting each of these sources using human judgment.

---

[6]https://www.qualtrics.com

[7]https://prolific.com

Given the size of the datasets, we implement this human-based sorting on a sample of each dataset. Since sorting large lists of elements all at once is quite cumbersome, we expand upon the method introduced in [9] to select pair-wise comparisons that can be presented to human participants. The method uses an efficient, stable sorting algorithm—TimSort—to select pairs for comparison. This way, it reduces the number of pairs that need to be presented to a human from $O(n^2)$ to $O(n \log n)$. It does this by supplying a custom comparison implementation to a standard sorting implementation that uses a database of collected pair-wise comparisons. If the order of a given pair is not yet known, the comparator throws an exception, breaking out of the sorting operation. The pair that was to be compared when the exception was thrown is then presented to the human participant for comparison.

The method in [9] assumes that a single person supplies all comparisons until a list of elements is sorted completely. Since we also want to be able to crowdsource the sorting task, we introduce several extensions to the method.

*Majority Voting.* To get a more consistent result when collecting comparisons from multiple humans, we use a majority voting scheme before accepting an order for a given pair. To do this, we simply present each pair to multiple participants until a pre-determined number agrees on an order.

*Sublist Sampling.* Since the next pair to be presented to a user is dependent on the previous sorting operations, waiting for a consensus on any particular pair would block all other participants using the original method. To overcome this issue, we introduce a sub-list sampling scheme, applying the sorting method not to the entire list, but to a randomly selected continuous sub-list. If a sub-list can be sorted without requiring any additional human input, we double the length of the sub-list and repeat the procedure until the sub-list length matches the length of the full list. Using such a sampling scheme, we avoid introducing a direct dependency between multiple human annotators. The downside of this approach is that it can lead to pairs being presented to an annotator that would not have been required if the full list were to be sorted using only the original method. Therefore, we also introduce another extension that can use information collected this way later in the process.

*Order Inference.* In the original method, the comparator would only check if the order for any specific given pair was already annotated by a human participant. Since the order of objects is a transitive property, we extend this by an inference mechanism. If the sorting algorithm requires the order for a pair $(A, B)$ for which the order is not yet known, our comparator will then look for a set of pairs $(A, X)$ and $(X, B)$ sharing a common third element. If such a combination of annotated pairs is found, the comparator will then infer the order for $(A, B)$ and add it to the database as an accepted order.

Since independent order annotations can lead to an inconsistent state, e.g., independent human annotators providing orders of the type $A < B$, $B < C$, $C < A$, violating transitivity, we also introduce a simple conflict resolution procedure. In case such a conflict is detected during the sorting operation, the pair that led to the conflict

is removed from the database of accepted ordered pairs and added to a dedicated list of inconsistent answers.

We provide the implementation of our human-in-the-loop sort as open-source software.[8] More details on the method are provided in the supplementary material.

## 5.3 Data sampling

For each of the five possible ground-truth sources identified in Section 4, we group all contained elements by their estimated semantic density and randomly select one element per group. This sampling is done because for any data source all elements with the same estimated semantic density are not distinguishable from each other. Since the ranges of estimated visual density vary from source to source, we end up with five lists of images with different lengths. Specifically, we select 104 images from Wikimedia Commons, 263 from YFCC100M, 112 from Visual Genome, 208 from LVIS, and 72 from the captioning datasets.

For each of the five sources, we sort the resulting list of selected images by the estimation mechanism applicable to the source. The resulting orders can then be compared to those generated by letting human participants sort the images by their perceived VSD. Comparing the respective orders gives us an estimate of how well each data source serves as a proxy for our measure.

## 5.4 Sorting experiment

To establish a baseline on the human perception of VSD, we perform an experiment in which we ask human participants to sort the lists obtained in Section 5.3 using the method described in Section 5.2. We sort each list in two different ways: First, each list is sorted by a *single* human annotator. (The list sampled from the image captioning dataset is independently sorted by two human annotators—two of the co-authors of this paper—to serve as an additional means of comparison.)

Second, we use Prolific to distribute the sorting task across 423 participants, each contributing with 50 pair-wise comparisons. Any pair needs to be sorted the same way by three crowd workers for its order to be accepted, ensuring that we will have a majority vote. The instructions provided to the crowd workers were phrased as follows: "*… we will show you several pairs of images. For each pair, please select the image that would take you more words to describe completely. When thinking of the description, imagine having to describe the image to an artist for them to draw it or to an archivist for them to find it in a large collection. You will not be asked to actually produce these descriptions, only to imagine their estimated relative lengths.*"

After sorting the lists based on human input, we compute the Spearman's rank correlation $\rho_s$ between the lists for each data source. Table 1 shows these correlations between the different orders obtained from the data source (*base*), the *single* human annotator, or the *crowd* generated order.

When comparing the orders generated by the two different human annotation-based methods, we see that there is moderate to strong agreement between the single annotator and the crowd, with correlation values ranging from 0.427 to 0.717. From this, we can conclude that there is a somewhat consistent human interpretation of VSD and that it can be at least roughly estimated without needing

---

[8]Link to repository removed for double-blind review.

**Table 1: Spearman rank correlation $\rho_s$ between 'base' ranking from the data source and the human-based sortings ('single' and 'crowd')**

| Data Source | Orderings | | $\rho_s$ |
|---|---|---|---|
| Commons | base | single | 0.365 |
| | base | crowd | 0.293 |
| | single | crowd | 0.717 |
| YFCC100M | base | single | -0.488 |
| | base | crowd | -0.358 |
| | single | crowd | 0.495 |
| LVIS | base | single | 0.030 |
| | base | crowd | 0.052 |
| | single | crowd | 0.427 |
| Visual Genome | base | single | 0.161 |
| | base | crowd | 0.165 |
| | single | crowd | 0.552 |
| Captions | base | single 1 | 0.562 |
| | base | single 2 | 0.538 |
| | base | crowd | 0.470 |
| | single 1 | crowd | 0.536 |
| | single 2 | crowd | 0.606 |
| | single 1 | single 2 | 0.652 |

actually to generate a complete image annotation. The alignment of the orders obtained from different annotators is, however, far from perfect, which can be attributed to the contextual dependency of the task. This gives us an upper bound of what can be reasonably expected when trying to estimate such semantic density values.

When comparing the orders obtained from the data sources with those generated by the different human annotators, we see a much larger range of results. For most data sources, the obtained correlation is weak or negligible, and in the case of YFCC100M, it is even considerably negative. Only the correlation between the estimate obtained from processing extensive image captions falls in a similar range compared to the correlations obtained by comparing different human annotations. It appears that extensive image captions serve as an adequate proxy for an image's semantic density.

## 6 CAPTION-BASED VSD ESTIMATION

Based on the results of the last section, we can see that values derived from the diversity of extensive image captions serve as a workable proxy for our notion of VSD, outperforming all other tested baselines. To test the stability of this approach, we next test if this is a property of the specific images used in these data sources, if this effect can be replicated for the images from the other sources as well, and if this can be done automatically. For this comparison, we automatically generate extensive image captions for all the images selected in Section 5.3 and derive a semantic density estimate from these captions in the same way as for Localized Narratives. We use the LLaVA [16] vision language model for caption generation and prompt it with an instruction similar to the one given to the crowd workers, as described in Section 5.4: *"Describe the image as if having to describe the image to an artist for them to draw it or to an archivist for them to find it in a large collection."* Afterward, we again sort the

list for each data source by the density estimate obtained this way and compare the rank correlation to the manually sorted baselines. The results are shown in Table 2.

When looking at the corrections between the orders obtained from these synthetic captions and those generated using both individual (*single*) and *crowd* annotations, we can see that all lists are positively correlated, with values ranging from 0.278 to 0.571. While this indicates that the effects observed in Section 5.4 are likely not an artifact of the image sampling and the caption-derived estimates are also applicable to other images, this synthetic captioning approach for VSD estimation has clear drawbacks. Not only is it computationally costly to run a large vision language model for caption generation, which potentially limits its applicability to larger collections, but the resulting orders, in some cases, only show a weak correlation with the manually obtained baselines. We, therefore, investigate if it is feasible to estimate the result of such a pipeline more directly and in a computationally more efficient manner.

## 7 END-TO-END VSD ESTIMATION

After identifying the values derived from extensive image captions as a reasonable proxy measure for VSD, we use it to train an end-to-end, automatic VSD estimation mechanism.

### 7.1 Method

To this end, we propose a simple neural network based on ConvNext [17]. We chose the ConvNext architecture due to its good trade-off between parameter count and classification performance. For our estimator, we start with a ConvNext tiny variant pre-trained on the ImageNet 22k classification task and replace the final classification layer with a dense layer with an output dimension of 2048 and a GELU [7] activation, followed by another dense layer with an output dimension of 1. The GELU activation is chosen to stay consistent with the architecture of the ConvNext model, which serves as a backbone. The estimator is then trained to predict the values obtained by the procedure described in Section 4.3 directly from the images, using an L1-loss.

As a training set, we use the train splits of all data sub-sets of Localized Narratives as well as the entirety of Stanford Paragraphs in combination. The test splits of Localized Narratives serve as a test set. To train the estimator, we first freeze the backbone and train the newly added layers for a single epoch with a learning rate of $10^{-2}$. Subsequently, the entire network is fine-tuned with a learning rate of $10^{-4}$ until the test loss stops decreasing. Analogously to the backbone model, the AdamW [18] optimizer is used for training, with its default values of $\beta_1 = 0.9$ and $\beta_2 = 0.999$.

We release our implementation and model weights as open-source software.[9]

### 7.2 Results

To evaluate the performance of our model, we use it to sort the test sets that were manually sorted in Section 5.4. To ensure comparability, none of the manually sorted images were part of the training set of the model. The resulting correlations between the

---

[9]Link to repository removed for double-blind review.

various baselines and the predicted order per data source are shown in Table 2.

The results show that the correlation between the different versions of the manually generated orders is similar (in range) to the correlation between any of the manually generated orders and the model's output order. The strongest correlation can be observed for the images from the Localized Narratives dataset. This is not surprising because the model was trained on the estimates generated from the captions. We can, therefore, see that the model can estimate the number of concepts extracted from extensive captions quite well without requiring the generation of the actual image description.

The results for the images obtained from YFCC100M and Visual Genome also show a substantial correlation with the human baseline. The correlation is still clearly positive, although somewhat weaker for the images taken from LVIS.

No correlation can be observed for the predicted order of the images from Wikimedia Commons. This might be because this dataset is the only one that is not exclusively composed of natural images but also contains artificial images, such as illustrations, paintings, etc. Since the model uses a backbone pre-trained on ImageNet and was then trained on images from Localized Narratives, it was only ever exposed to photographs of natural scenes, which might explain its poor performance on the visual content of a different type. To test this hypothesis, we manually exclude all non-natural images from the lists and re-compute Spearman's $\rho_s$ for only the remaining natural images. The results are also shown in Table 2 as 'Commons (Natural).' Examples of natural (i.e., photographs) and non-natural images are shown in the supplementary materials.

As we can see, the correlations increase substantially, confirming our hypothesis that the model is limited to natural images and does not generalize to all types of visual content. For the images taken from YFCC100M, Visual Genome, and Localized Narratives, the correlation between the human-annotated order (single and crowd) and the one obtained via the end-to-end prediction is clearly larger than the one obtained using synthetic captions. The same is true for the natural images from Wikimedia Commons, although not for the complete Commons sample, as discussed above. Only for the LVIS sample, the order predicted using the synthetic captions is more aligned with the human annotation. The difference between the two methods on LVIS is, however, comparatively small.

## 8  DISCUSSION

After having investigated multiple ways of estimating VSD, this section tries to synthesize the insights that can be gleaned from the results presented in Tables 1 and 2. As discussed previously, it is presumably not generally possible to determine a precise and context-independent VSD value for any given image. Since we do not want to limit our investigation to any specific context, we relax the conditions slightly and only consider the order that a collection of a collection of images would have when sorted by this semantic density value. This is sufficient for our ultimate goal of identifying gaps in annotations by comparing visual documents in a given collection since images with similar semantic density are expected to have a similar number of annotations.

**Table 2: Spearman's $\rho_s$ between manually sorted images and predicted image order based on synthetic image captions and end-to-end VSD prediction.**

| Data Source | Order | $\rho_s$ synth. captions | $\rho_s$ end-to-end |
|---|---|---|---|
| Commons | single | 0.278 | -0.023 |
|  | crowd | 0.418 | 0.010 |
|  | base | 0.303 | 0.102 |
| Commons (Natural) | single | 0.210 | 0.458 |
|  | crowd | 0.438 | 0.548 |
|  | base | 0.355 | 0.396 |
| YFCC100M | single | 0.495 | 0.695 |
|  | crowd | 0.499 | 0.552 |
|  | base | -0.350 | -0.526 |
| LVIS | single | 0.472 | 0.439 |
|  | crowd | 0.375 | 0.316 |
|  | base | 0.125 | -0.069 |
| Visual Genome | single | 0.321 | 0.507 |
|  | crowd | 0.402 | 0.561 |
|  | base | 0.199 | 0.105 |
| Captions | single 1 | 0.565 | 0.706 |
|  | single 2 | 0.571 | 0.637 |
|  | crowd | 0.408 | 0.641 |
|  | base | 0.436 | 0.585 |

Based on our definition of VSD given in Section 3, we collect several image datasets with accompanying annotations and established means for their generation that we suspect could serve as a proxy for our measure. Sampling these datasets gives us a test collection that we can then sort based on the semantic density estimates obtained from the respective datasets. We also set up a human-in-the-loop sorting pipeline to manually sort the same images, using both a single annotator and the aggregation of multiple crowdworkers. Based on these results that were presented in Table 1, we can summarize several insights:

When comparing the orders obtained by the two different manual sorting mechanisms, we get a range of Spearman's $\rho_s$ from 0.427 to 0.717, with a median correlation value of 0.552. These results tell us that *while there is a clear positive correlation between the estimate of our individual annotators and the aggregate of many crowd workers, they are by no means in perfect agreement*. The range of agreements between manual annotations also serves to ground the correlations obtained by other means.

Looking at different options for proxy measures for our notion of VSD, based on established datasets used by the multimedia and computer vision communities for various applications, we can see that *only estimates derived from extensive image captions produce VSD-based orders with a correlation to the human estimate in a similar range as the values discussed above.* When generating synthetic image captions by promoting a state-of-the-art vision language model and subsequently processing these captions in the same way as the manually generated ones, we can obtain image orders with a consistently positive, although not very strong, correlation to the manual orders, as shown in Table 2. This shows that even *synthetic captions can serve as a proxy measure for VSD, but the quality of the*

*estimates is rather limited.* Additionally, the computational cost of this approach is very high.

We achieve much *higher correlations with the manual baseline by adapting a state-of-the-art image classification model* by replacing its final classification head with a simple regression head and training it on VSD estimates obtained from manually generated, extensive image captions. This approach obtains correlation values ranging from 0.316 to 0.716 with a median of 0.552, achieving a range comparable to the agreement between the different manual baselines, see Table 2. This approach consistently outperforms the synthetic caption-based approach while being substantially computationally cheaper. It is, however, limited to natural images due to the composition of the training set.

Figure 2 illustrates the image orders generated by the different methods. Samples of the actual image sequences are shown in the supplementary materials. The figure uses the *crowd* order as a basis to assign increasing color intensity values to the positions in the series. It uses different colors for the other ordering methods. The more a horizontal line of one color resembles a continuous gradient, the higher the agreement between the *crowd* order and the method. The figure shows that, while there is no clear continuous gradient other than the blue reference, the green lines representing our end-to-end prediction mechanism consistently place high-intensity elements to the right and low-intensity elements to the left. The same is not true for the grey lines (i.e., the base), which generally do not resemble any gradient. When looking at the violet lines, representing the order generated based on the annotations of a single annotator, we can see two gradients rather than one. This could be an artifact of our sorting mechanism, caused by only a small number of pairs that were annotated differently between the single and the crowd annotations. These differences can cause differences in the merge behavior of the sorting algorithm, causing two sub-lists to be merged differently. In view of the uncertain nature of the human interpretation of the measure, future work might include considering increasing the robustness against small perturbations in the pairwise comparisons.

The obtained results show that *it is feasible to automatically sort images by an estimate of their VSD and achieve results comparable to a human baseline.* Based on this, it is reasonable to assume that *such an automatically derived measure could be used for estimating the annotation completeness* of visual media in digital collections and archives, such as Wikimedia Commons, by comparing the number of annotations per document with their relative semantic density. *A large discrepancy between the per-document ratio of these values can be an indicator of missing annotations.*

## 9 CONCLUSION AND FUTURE WORK

In this paper, we introduced the notion of VSD, a measure of the amount of visual information contained within an image. We showed empirically that this measure could be approximated by a manual image sorting task, but it can not be determined precisely this way. Given its context dependency, it is unclear what level of precision is even possible for a general measure. We were able to show that the linguistic content and diversity of extensive image captions serve as a suitable proxy measure for VSD and that this also holds to a reduced degree for synthetic captions. With a

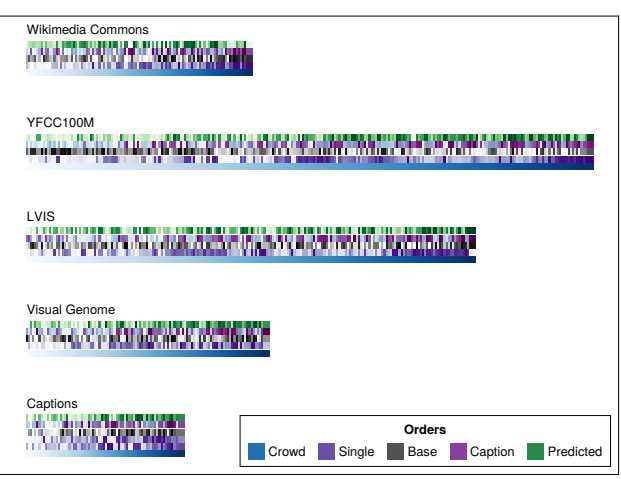

**Figure 2: Visualization of the different image orders as generated by different methods. The order defined by the crowd is used as a basis for each dataset. Colors show different ordering methods. Intensity shows the position of an image in the list ordered by the crowd. The more a line resembles an uninterrupted gradient, the more aligned it is with the crowd order.**

ground truth dataset derived from detailed manual image captions, we could train a model that estimates the VSD of a given image without the need for caption generation. The order of images sorted by this estimate shows correlation values comparable to lists of images manually sorted by independent annotators.

The primary motivation for this work was to create a completeness estimator for the annotations of visual media in digital collections and archives. Similar to completeness estimation methods employed for other modalities in collaborative knowledge repositories such as Wikidata, such a method could be used to help the community identify and close gaps in annotations or interrelations. By comparing the amount of already existing annotations across a collection with the predicted VSD per image, the images that are under-annotated with respect to others can be identified.

So far, we have not evaluated the perceived usefulness of our measure and the impact on the behavior of human annotators completing image annotations, but we aim to do so in future work. Other possible extensions of this work include investigating the generalization from static images to video or the applicability to different modalities such as audio. Finally, even though this paper was limited to one usage scenario for VSD—around image description completeness—we believe this measure could be helpful for other multimedia tasks to measure the information density of an object.

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
