# OpenReview forum: "Estimating the Semantic Density of Visual Media"
_acmmm.org/ACMMM/2024/Conference — MM2024 Poster_

### Official Review · Reviewer_Qxnr · 2024-05-23

**Rating:** 4
**Confidence:** 3

**Summary:**

The paper introduces the concept of visual semantic density for images, a measure to assess how much semantic information is contained in an image. Based on that the authors conduct a couple of experiments to on the one hand examine the concept and on the other hand check how good the concept is correlating to different annotation approaches (keywords to automatic caption generation). Finally models are trained to directly estimate the measure from an image. For evaluation a number of image datasets (with annotation sis used).

**Strengths:**

* Measures as the one introduced are highly needed as measures which have also higher-level meaning to humans are scarce (although the measure is defined rather simply and also is relative between two images, this has the advantage that the concept could be grasped quite rapidly).
* The sequence of experiments underlines that the concept is valid and that correlation between the human understanding of the concept and the learned end-to-end model is good.
* The software and the model weights are released under an open regime.

**Limitations:**

* The paper only addresses images (no other modalities).
* The presentation, although overall good, could be improved (see below).

### Suggestions
* Section 2:  Segmentation is mentioned twice once in regards to [10] and in another paragraph in regards to [14]. This makes it hard for the reader to comprehend and it is also not obvious why the same approach is mentioned in different places. So it is suggested to consolidate this.
* Table 2: The experiments should be presented in a similar order (base – single – crowd) as in Tab 1

### Questions
* Linen182: what is meant with “the approach still depends on the input text” (e.g., does this  refer to the input text used for CLIP learning)?
* Fig 2: Rationale for using crowd correlation and not single correlation - having the highest correlation values (it seems to be better arguable that a single person is consistent in the judgements than a crowd)?

### Wording/Typos
* Line 101: “ Our end goal“ -> “ Our _ultimate_ goal“
* Line 128: “manual” and compute in “the manual orders that we compute” seem to be contradicting, furthermore instead of “orders” probably “rankings” is intended
* Line 148f: “synthetic image captions or and end-to-end approach” -> “synthetic image captions or _an_ end-to-end approach,”
* Line 287: “will be at most of the sum” -> “will be at _most the_ sum”
* Line 748f: “order that a collection of a collection of images”-> “order _that a_ collection of images”

**Suitability:**

2

---

### Official Review · Reviewer_hvRQ · 2024-05-24

**Rating:** 6
**Confidence:** 4

**Summary:**

The paper introduces the concept of visual semantic density, defined as the information required to comprehensively describe an image so its content can be accurately inferred from the description.For that research the authors conducted studies with users to find an objective measure, created an estimator and evaluated their approach.

**Strengths:**

This paper is self-contained, easy to read and well-written. The authors explain the basics in short summarizes, bring the reader up to speed and then then explain their approach and findings. As this is interdisciplinary work, combining methods and models from psychology and computer science this is necessary and very well done. The topic is IMHO very important. Many researchers and practicitioners take annotations for granted and have no objective measures on their quality or completeness. The paper fills a real gap here.

**Limitations:**

Please do not take my comments as limitations of the work presented in the paper, but more like as a next step.
(1) In my research experience, it makes sense to closely investigate the context of producing, organizing and consuming visual media. Examples are the sharing of experiences and emotions, using images as a social communication medium, or of course self-expression.
(2) I recommend to take a look at image descriptions for accessibility. There people definitely have thought about how much text is needed to convey what the image is expressing, and how this is different to your use case.
(3) I'd also turn it around and use the generated image description to create images with a text-to-image model like Stable Diffusion or GPT. The original and the generated image can then evaluated and one could hopefully find indications how the "completeness" of such a description impacts the difference between original and generated image.

**Suitability:**

3

---

### Official Review · Reviewer_CyfK · 2024-05-31

**Rating:** 3
**Confidence:** 2

**Summary:**

The paper explores an interesting and import topic, which is very challenging because it is difficult to measure the semantic density of the visual media. While the paper is well written with good structure and presentation, it is not clear how the proposed method is a good way to measure the semantic density, and how the result of the measurement could be used to improve the image annotation tasks.

**Strengths:**

The paper explores a very interesting topic that aims to improve the quality, efficiency, and effectiveness of the image annotation. The paper re-defined visual semantic density (VSD) proposed a method to estimate the optimal VSD for an image. The method is evaluated by six date sets using the Spearman rank correlation between human judgement and the model prediction.

**Limitations:**

While the paper explores an important topic, there is a lack of details and justification on evaluation/experiment set up and results. It is unclear how the results demonstrate the effectiveness of the method and why? How the proposed method could be applied in most of situations and on other datasets. It would be helpful to show the significant test result on the results.

**Suitability:**

2

---

### Meta-Review · Area_Chair_Z5Qp · 2024-07-03

**Recommendation:** Accept (Poster)
**Confidence:** 4

**Metareview:**

The paper presents an innovative approach to measuring visual semantic density, which has the potential to significantly impact the field of multimedia and image annotation. While Reviewer CyfK raised concerns regarding the evaluation setup and practical applicability, the rebuttal provided by the authors offers reasonable clarifications and improvements. Reviewers hvRQ and Qxnr found the concept highly valuable, with the latter providing constructive suggestions for enhancing the paper’s presentation.

I recommend acceptance of this paper, given the topic’s importance and potential impact on multimedia processing and image annotation.
The paper has an overall positive reception by the majority of reviewers, that highlights its well-written and interdisciplinary nature, stating that it addresses a gap in objective measures for annotation quality and completeness.